# Cryostructuring of Polymeric Systems: 64. Preparation and Properties of Poly(vinyl alcohol)-Based Cryogels Loaded with Antimicrobial Drugs and Assessment of the Potential of Such Gel Materials to Perform as Gel Implants for the Treatment of Infected Wounds [note 1]

**DOI:** 10.3390/gels9020113

**Published:** 2023-01-28

**Authors:** Olga Yu. Kolosova, Astemir I. Shaikhaliev, Mikhail S. Krasnov, Ivan M. Bondar, Egor V. Sidorskii, Elena V. Sorokina, Vladimir I. Lozinsky

**Affiliations:** 1A.N.Nesmeyanov Institute of Organoelement Compounds, Russian Academy of Sciences, Vavilov Street 28, Bld. 1, 119334 Moscow, Russia; 2Institute of Dentistry, I.M.Sechenov First Moscow State Medical University (Sechenov University), 119991 Moscow, Russia; 3Microbiology Department, Biology Faculty, M.V.Lomonosov Moscow State University, 119991 Moscow, Russia; 4Microbiology Department, Kazan (Volga-Region) Federal University, 420008 Kazan, Russia

**Keywords:** poly(vinyl alcohol) cryogel, Ceftriaxone, Fluconazole, drug release, antimicrobial implants, in vitro and in vivo bioassay

## Abstract

Physical macroporous poly(vinyl alcohol)-based cryogels formed by the freeze–thaw technique without the use of any foreign cross-linkers are of significant interests for biomedical applications. In the present study, such gel materials loaded with the antimicrobial substances were prepared and their physicochemical properties were evaluated followed by an assessment of their potential to serve as drug carriers that can be used as implants for the treatment of infected wounds. The antibiotic Ceftriaxone and the antimycotic Fluconazole were used as antimicrobial agents. It was shown that the Ceftriaxone additives caused the up-swelling effects with respect to the cryogel matrix and some decrease in its heat endurance but did not result in a substantial change in the gel strength. With that, the drug release from the cryogel vehicle occurred without any diffusion restrictions, which was demonstrated by both the spectrophotometric recording and the microbiological agar diffusion technique. In turn, the in vivo biotesting of such drug-loaded cryogels also showed that these materials were able to function as rather efficient antimicrobial implants injected in the artificially infected model wounds of laboratory rabbits. These results confirmed the promising biomedical potential of similar implants.

## 1. Introduction

Poly(vinyl alcohol)-based cryogels (**PVACGs**) are the physical (non-covalent) macroporous gel materials that are formed as a result of cryogenic processing (freezing, incubation in a frozen state, and thawing) of concentrated aqueous or DMSO solutions of the polymer [1,2,3,4,5,6,7,8,9,10]. Apart from substantial scientific interest, PVACGs have a wide application potential, especially in such fields as biotechnology [3,5,9,10,11,12,13,14,15,16,17,18,19] and medicine [1,3,5,9,10,20,21,22,23,24,25,26,27,28,29,30,31,32,33,34,35,36,37,38,39,40,41,42,43,44]. Since there are numerous possibilities to vary the physico-mechanical and thermal (heat endurance) properties of PVA cryogels, as well as their macroporous morphology, these parameters can be precisely adjusted to the characteristics that are required for each particular applied task. Such processing variables are as follows: the molecular weight of the used PVA, its deacylation degree, the tacticity of chains, polymer concentration in the feed solutions, the employed solvent, the presence of soluble or insoluble (fillers) additives, the temperatures/time regimes of the cryogenic processing stages, and the number of the freeze/thaw cycles [1,2,3,4,5,6,7,8,9,10,45,46,47,48,49,50,51,52,53,54,55,56,57,58,59,60,61,62,63,64,65,66,67]. The influence of these variables on the macroporous morphology of the resultant PVA cryogels has also been explored and reported in detail [3,5,6,7,8,9,10,47,48,53,54,55,60,61,62,63].

As for the biomedical application of PVACGs, many examples have been described, including the implementation of these materials as drug delivery vehicles [4,26,27,38,41,44,65,68,69,70], wound dressings [22,71,72,73,74,75], prostheses of cartilages [4,20,23,28,29,30,32,34,76,77] and heart valves [25,40], and the use of these cryogels as so-called phantoms (gel standards and models) for medical NMR tomography [21,78,79] and ultrasound diagnostics [24,80,81].

In this context, the PVACGs capable of performing as drug delivery implants, especially those carrying antimicrobial medications, are of significant interest for dentistry and maxillofacial surgery. This is because strongly infected wounds and abscesses are the cases rather frequently met in the respective medical practices [82]. One of the promising approaches to the treatment of such lesions (deep wounds, in particular) is the use of the drug-bearing hydrogel-based temporary implants that, on the one hand, are able to release necessary medication directly into the infected zone, and, on the other hand, are capable of preventing the uncontrolled fusion of disturbed tissues. After completion of the antimicrobial treatment, such implant is gently removed from the sanitized wound, which is further subjected to standard surgery processing. It is this approach that we have recently exploited for the fabrication of temporary implants based on the PVACGs that contained various antibiotics [83]. Thanks to the macroporous morphology of PVA cryogels, the drug release from the respective medication-loaded PVACG vehicles does not, as a rule, have serious restrictions [3,4,5,27,28,29,44,68,74]. This quality has also been shown to be characteristic of the PVA cryogels that contain various entrapped antibiotics, for instance tylosin tartrate and oxytetracycline hydrochloride [84], enrofloxacin [85], oxytetracycline [86], ampicillin [87], and some others. In the present study, the PVACG-based implants also contained antimicrobial additives, namely, the antibiotic substance Ceftriaxone (**CFT**) (Figure 1a) and, for the sake of antifungal defense in the case of in vivo experiments, the antimycotic agent Fluconazole (**FNZ**) (Figure 1b). CFT relates to the cephalosporine family and possesses a rather broad antibacterial activity; FNZ exhibits pronounced fungicide action [88].

The goals of this our research were the evaluation of the physicochemical properties of the prepared CFT-bearing PVA cryogels and the biotesting of CFT/FNZ-PVACGs in the course of in vivo experiments in order to confirm the biomedical potential of similar drug delivery materials. To the best of our knowledge, the PVACG-based implants of the above-indicated composition were previously unknown. Respectively, such CFT- and CFT/FNZ-containing gel materials are the novel ones; the peculiarities of the procedure for their loading with the antimicrobial drugs, the physicochemical properties of the resultant implants, and their suitability for the surgical implementation have been developed and investigated for the first time.

## 2. Results and Discussion

### 2.1. Preparation and Physicochemical Properties of the Drug-Free and Drug-Loaded PVA Cryogels

In this study, the drug-free PVA cryogels have been prepared (Section 4.2) either in the shape of cylinders (Figure 2a) using sectional duralumin molds (inner dia. 15 mm, height 10 mm), or as the 3 mm thick discs (Figure 2b,c) molded in the plastic Petri dishes (inner dia. 40 or 60 mm). Thereafter, cylindrical samples were used in the physico-chemical experiments, and the “flat” PVACGs were employed for the biological tests, when the samples of the required size were cut from the respective discs using sharp medical scissors.

Since there are two principal approaches to the preparation of drug-load ed gel materials [91], the following options were considered in the case of drug-carrying PVA cryogels:(i)The first way is the incorporation of desired medications in the precursor PVA solution followed by its gelation, thus entrapping the target substances in the resultant gel carrier. It is this same method as the preparation of the antibiotic-carrying PVA cryogels used in the known studies [83,84,85,86,87] mentioned in ‘Introduction’.(ii)The second approach is the initial preparation of the drug-free gel matter, its rinsing, when it is required, from the possible admixtures, and then drug uploading into the gel by its immersion and incubation in the drug-containing solution for the saturation of the carrier with target substances.

Although the second way is somewhat more laborious and longer, it has certain advantages. One of those is the possibility “to launder” the gel carrier prior to its loading with a drug. This action is of special importance in the case of gels formed via the polymerization of low-molecular monomers, some of which are often rather toxic. Another point relates to the storage possibilities of the resultant implant prior to its surgery application. If the loaded medication possesses limited hydrolytic stability, it is preferable to incorporate the drug into the hydrogel carrier virtually ahead of the medical use, i.e., the time interval between the drug incorporation into the implant and the insertion of the latter into the patient’s body should be as short as possible. In turn, the stable drug-free gel carrier can be stored for a long time.

Based on this reasoning, the approach (**ii**) has been chosen in the present study to model the real procedure for the preparation of target implants. In the preliminary experiments it was found that at room temperature, the time of the CFT loading into the PVACG cylindrical samples (Section 4.3) was about 72 h in order to reach the equilibrium of the drug concentration inside the gel carrier and in the outer CFT solution. It is clear that such loading duration is defined by the diffusion rate and, therefore, will vary depending on the geometry parameters of the PVACGs to be saturated with the drug. Within the framework of this study, we used 72 h downloading in all cases, i.e., for both the gel cylinders and the gel discs (Figure 2).

It was initially required to answer the following question: can the physical characteristics of PVACGs deteriorate due to the gel matter loading with CFT?

This question was stipulated, on the one hand, by the known influence of various solutes on the swelling behavior, rigidity, and heat endurance of PVA cryogels [56,61,62,63,64,65,66,74,87], and, on the other hand, by the potential suitability or unsuitability of the physical properties of the CFT-loaded PVACGs for their implementation as gel implants. Thus, an example of an undesirable effect would be the excessive up-swelling and strong softening of a PVACG-based implant after its loading with a drug. Since the main mechanism responsible for the PVA cryotropic gelation is the interchain hydrogen bonding [3,4,5,6,92,93,94], the substances capable of preventing the H-bonding, i.e., acting as the chaotropes such as urea or guanidine hydrochloride [61], can “worsen” the medical applicability of the respective implants to a greater or lesser extent. The graphs in Figure 3 allow for a comparison of the physico-chemical characteristics for the drug-free and CFT-loaded PVACGs.

It was found that the volume of PVACG cylindrical samples increased by a factor of ~1.72 (Figure 3a) after their equilibration with the 0.2 M CFT aqueous solution, i.e., PVA cryogels additionally swelled upon loading with CFT. This fact points to the manifestation of certain chaotropic properties inherent in Ceftriaxone. In turn, the Young’s moduli for the drug-free and CFT-loaded cryogels differed insignificantly (Figure 3b), these *E*_c_ values were, respectively, 9.7 ± 0.50 and 10.1 ± 0.3 kPa, i.e., the differences were within the experimental error. Such results can testify to a positive contribution of the up-swelling pressure to the elasticity characteristics of the CFT-loaded PVACGs (Figure 3a), thus compensating for some swelling-induced softening of the gel matrix. Of course, additional studies are required to confirm or to refute this assumption. However, similar studies were not included in the objectives of our present research. Finally, the fusion temperature (*T*_f_) of the latter samples markedly decreased to 64.5 ± 0.3 °C in comparison with *T*_f_ = 74.0 ± 0.1 °C for the drug-free PVA cryogels (Figure 3c). Since it is well known that the heat endurance of physical hydrogels such as those based on gelatin [95] or agarose [96], and PVACGs in particular [3,4,5,6,92,93,94], is determined by the amount of interchain hydrogen bonds in the knots of polymeric 3D networks, the data of Figure 3c indicate a partial breaking of such bonds as a consequence of CFT chaotropic influence. With that, the number of the intermolecular cross-links decreases, thus resulting in an increase in the length of the mobile chain segments and, hence, in the growth of gel swelling (Figure 3a). Nonetheless, the experimental results that are shown in the graphs of Figure 3 allow us to draw an evident conclusion that the loading of a PVA cryogel with CFT did not cause such a deterioration in the physical properties of the gel carrier that could significantly interfere with the application of a similar drug-loaded material as an implant for the treatment of infected wounds.

As for the drug-containing PVACGs loaded with a mixture of CFT + FNZ, it was found that the FNZ additive at the concentration used in this research virtually did not affect the *V*_s_, *E*_c_ and *T*_f_ values of the respective cryogels in comparison to the same parameters inherent in the PVACGs containing only CFT (Figure 3). Therefore, the data for the CFT/FNZ-PVACGs are not given in these plots.

### 2.2. CFT Release from the Drug-Loaded PVA Cryogels

The features of the GFT release from the drug-carrying PVACGs were evaluated in two ways:

(A) Simple diffusion of the drug from the cryogel carrier to the neat water (Section 4.3);

(B) Release of CFT from the drug-bearing carrier to the microbial mat for the evaluation of the bactericide action of the system PVACG/CFT with respect to different microbial strains (Section 4.4).

In the former case, the known Weibull’s equation [97,98] was used for the evaluation of the release kinetics (Figure 4); this formula is as follows:*M*_t_/*M*_∞_ = 1 − exp(−*a* × *t^b^*)(1)
where *M*_t_/*M*_∞_ is the solute fraction released from the matrix for time ***t***; the values of ***a*** and ***b*** constants were obtained using the ORIGIN PRO software (OriginLab Corp., Northampton, USA) by uploading the Weibull’s equation and the experimental data into this program. In this equation, the constant ***b*** is a descriptor related to the influence of the gel matrix structure on the drug release, and the ***b*** value equal to 0.846 found in this study for the PVACG/CFT system should, according to [99], testify to the free diffusion mechanism for the release of ceftriaxone molecules from the macroporous matrix of a PVA cryogel. In other words, this means that CFT is predominately located inside the space of macropores.

In this context, the non-hindered CFT release from the PVACG-based carriers has also been observed upon the evaluation of their bactericide activity, which was tested by the agar diffusion technique (Section 4.5). The images of the respective agar dishes (Figure 5) exemplify the so-called “growth inhibition zone” (**GIZ**) [100] formed around the PVACG/CFT samples located onto the microbial “lawns” of three different bacterium strains. The quantitative data on the GIZ diameters for these systems are summed up in Table 1.

When the bacteria under study were sensitive to CFT, no areas of culture growth were formed around the cryogel sample. The growth inhibition zone of more than 30 mm (Table 1) indicated a high sensitivity of the microorganisms (*St. aureus* and *E. coli*—Figure 5a,b, respectively) to Ceftriaxone, whereas low sensitivity of the Ps. Fluorescens cells (Figure 5c) to CFT is explained by the fact that not all strains of such species are sensitive to this antibiotic [88,89]. Therefore, in spite of the efficient CFT release from the drug-loaded PVA cryogel (Figure 4), the GIZ values in the case of the latter cells were relatively low (Table 1).

### 2.3. In Vivo Experiments

The final stage of our study concerned the assessment of the potential of CFT/FNZ-bearing PVACGs to act as antimicrobial implants suitable for the healing of infected wounds. The respective in vivo experiments were carried out by the subcutaneous implantation of such cryogels (the samples were of 1.0 × 1.0 × 0.5 cm size in all cases) into the artificially infected model defects in the thighs of laboratory rabbits (groups 1 and 5, Section 4.6) and into the none-infected thighs (reference examples—group 4, Section 4.6); whereas the infected animals without implantation (group 2, Section 4.6) were used for the sake of comparison. Although we mainly consider such drug-loaded PVA cryogels as removable, i.e., temporary implants, in the present research the implanted CFT/FNZ-PVACG samples were not taken from the rabbit bodies, since it was necessary to prepare the histological sections of the tight tissue areas in direct contact with the surfaces of a gel implant.

As a result, the following features of the histological patterns (Figure 6) have been revealed:

Group 1—the rabbits withdrawn from the experiment 7 days after infection and injection of the CFT/FNZ-containing implants. In this case, i.e., at a relatively early stage, a manifestation of inflammation was observed (Figure 6a), namely, the formation of fibrous capsules (1) around the cryogel and the presence of a focus of dead leukocytes in the form of pus outside the capsule (2). The beginning of the germination of blood vessels into the area of damage (3) is also seen. The latter fact may indicate an earlier process of granulation and disinfection of the wound from the microbes.

Group 2—the rabbits removed from the experiment 7 days after infecting and suturing the wound without the insertion of CFT-carrying implants. For these animals, a pronounced picture of inflammation (Figure 6b) in the area of infection was observed as purulent-necrotic processes (4) with micro abscesses (5). In turn, neither fibrous capsule limiting the area of inflammation, nor vascular germination into the area of inflammation, were found.

Group 3—the animals operated similarly to the rabbits of group 1, but withdrawn from the experiment on the 14th day (Figure 6c). Existing inflammation, but with a reduced number of bacterial colonies in the area of infection, was observed near the fibrous capsule outside the implant. The pus (6) included dead cells of the immune system and, cells of microorganisms were observed around the cryogel. There was also the appearance of granulation tissue in the area of damage. A demarcation zone (7) was formed at the border of inflammation, and collagen fibers (8) and fibroblasts (9) were formed, which are shown at a higher magnification in Figure 6c’.

Group 4—the non-infected rabbits withdrawn from the experiment 30 days after injecting the CFT-containing implant. In this case, as it was expected, no inflammation in the area of non-infected wound with inserted implant was detected (Figure 6d). With that a fibrous capsule (10) has been formed around the gel implant, and some germination of blood vessels (11) was observed. Such a pattern is typical for healing of the non-infected wounds in the presence of a foreign insoluble insertion.

Group 5—the animals operated similarly to the rabbits of group 1 but were withdrawn from the experiment on the 30th day. In fact, the data for this group of laboratory animals allowed us to draw an “optimistic” conclusion on the suitability of the drug-loaded PVACGs to be used as implants for the healing of infected wounds. First, in the respective histological section (Figure 6d), the formation of the fibrous capsule (12) around the cryogel-based implant was observed. Second, in the area of a slight inflammation (13), obviously owing to a reduced sensitivity of some minor microbes in the composition of infecting consortium to the CFT action, no traces of pus were found in the wound area, thus indicating that on the 30th day, most of the bacteria introduced into the wound with droppings were almost completely suppressed. With that, the bulk of the tissue was occupied by mature collagen fibers (14).

As for the comparison of data for the animals of groups 3 and 5 with the animals of control group 2 (the infected rabbits without the drug-loaded implants withdrawn from the experiment on day 7), similar control rabbits died before the 14th day as a consequence of a strong infectious effect. Therefore, for the 14- and 30-day animals of groups 3 and 5 (initially infected rabbits with inserted drug-containing implants), comparison with the control was not possible.

Therefore, summing up the data obtained with respect to the effect caused by the CFT/FNZ-PVACG implant introduced into the infected wound, we can make a preliminary conclusion that the release of antibiotics from the cryogel matrix resulted in the destruction of bacteria in the closed wound, promoted the processes of wound cleansing and granulation, as well as lead to the recovery of the wound during such experiments. In turn, for the non-infected wounds, it was shown that the introduction of foreign material in the form of antibiotic/antimycotic-loaded PVA cryogel did not cause inflammation for 30 days, and a fibrous capsule was formed, thus isolating the implant from the surrounding connective tissue in the skin.

These conclusions have also been confirmed by results of microbiological analyses of the smears taken from the wounds of tested rabbits on the 30th day and sowing on microflora. If the smear from the infection source (rabbit droppings) gave about 24,000 CFU in 1 mL of the 48 h cultural liquid, the same parameters for smears taken from the 30 d wounds that were initially infected and injected with the CFT/FNZ-containing implants turned out to be 100–150 CFU/mL. The latter values are the normal microflora status for the tissues of the health rabbits. With that, no fungal contamination was found, thus indicating the antimycotic efficacy of the FNZ additives released from such implants, whereas the smears from the wounds of infected rabbits without the implants (reference group 2) contained fungal admixtures.

## 3. Conclusions

Various biocompatible and non-toxic PVA-based non-covalent cryogels that are formed via the technologically very simple freeze–thaw procedure are now proposed and used in biomedical fields. In particular, these rubber-elastic macroporous gel materials possess a set of properties valuable from the point of view of the PVA cryogels’ application in surgery. One example is the drug delivery matrices capable of functioning as implants, including temporary ones, carrying antimicrobial medications for the treatment of infected wounds. In this regard, in this study, similar PVACG-based model templates were prepared, loaded with antimicrobial drugs (CFT or CFT/FNZ blend) and examined upon in vitro and in vivo biological tests. It was found that the saturation of neat PVA cryogels with 0.2 M aqueous CFT solution resulted in a ~1.72-fold growth of the volume of the gel samples and caused a decrease in the fusion temperature of the matrix by about ten degrees Centigrade (from ~74 to ~64 °C), i.e., it led to a certain deterioration in the heat endurance of the polymeric material. These facts testify to the chaotropic properties of CFT, i.e., to its ability to break some interchain PVA-PVA H-bonds within a physical 3D -network of PVACGs. At the same time, the Young’s moduli for the drug-free and CFT-loaded cryogels differed insignificantly, i.e., loading of the PVA cryogel with CFT did not cause such a deterioration in the physical properties of the gel carrier that could significantly interfere with the application of similar drug-loaded materials as implants for the treatment of infected wounds. The CFT release from the cryogel vehicle was examined spectrophotometrically and analyzed using the Weibull equation. It was shown that the release of this drug occurred without any diffusion restrictions. In addition, the microbiological agar diffusion technique also demonstrated an efficient release of CFT from the drug-containing cryogel and, as a consequence, the suppression action of the antibiotic against the respective bacterial strains. The in vivo testing of the CFT/FNX-loaded cryogels was performed using the laboratory rabbits with the artificially infected model wounds. The histological and microbiological analyses of the wound tissue samples taken from the animals withdrawn from the experiment on the 7th, 14th, and 30th days after surgery operation revealed virtually a complete disappearance of infection by the 30th day. These results evidently testify that antibiotic/antimycotic-loaded PVA cryogels significantly accelerate wound cleansing and reduce the stage of inflammation, contributing to the earlier and active development of the proliferation stage in the treatment of purulent wounds. In other words, similar implants are able to act as quite efficient antimicrobials, thus demonstrating the promising biomedical potential of such drug-carrying gel materials. In the nearest future, we plan to examine the respective drug-loaded PVACG-based matrices as temporary implants with the separate impact on the tissue healing after the removal of the respective gel “insertions”.

## 4. Materials and Methods

### 4.1. Chemicals

The following substances and reagents were used in the experiments without additional purification: approved for biomedical use Mowiol^®^ 28-99 poly(vinyl alcohol) (**PVA**) (molecular weight of ca. 145 kDa, the deacetylation degree of 99%) was from Merck KGaA (Darmstadt, Germany), the antibiotic Ceftriaxone (**CFT**) and the injection quality water were from Rafarma Lld. (Lipetsk region, Russian Federation), the antimycotic Fluconazole (**FNZ**) was from Biocom (Stavropol, Russian Federation). GMF-agar microbiological medium was purchased from NITsF Ltd. (Saint-Petersburg, Russian Federation). The anesthetics for surgery manipulations were Zoletil 100 (Virbac S.A., Carros, France) and diethyl ether (Lenreaktiv, Saint-Petersburg, Russian Federation). Substances and consumables for histology were as follows: formalin, paraffin, hematoxyline, eosine (all BioVitrum, Saint-Petersburg, Russian Federation), ethanol (Ferrain, Moscow, Russian Federation), acetone and xylene (Panreac, Barcelona, Spain), microscope slides (Epredia, Braunschweig, Germany), cover glasses (Assistant, Sondheim, Germany).

### 4.2. Preparation of PVA Cryogels

The PVACGs were formed essentially in accordance with the earlier published procedure [54,55,61] with minor modifications. Briefly, 10 g of PVA was dispersed in 100 mL of water, kept for swelling at room temperature overnight, then heated by stirring on a boiling water bath until the complete dissolution of the polymer and then cooled down to a room temperature. The resultant solution was weighed before and after heating, and the amount of evaporated water was compensated, thus giving rise to the aqueous PVA solution of the 100 g/L concentration. Such solution was then poured into the necessary molds that were placed into the chamber of a precision-programmable cryostat Proline 1840 (Lauda, Königshofen, Germany), where the samples were frozen, incubated at −20 °C for 12 h, and defrosted by raising the temperature to 20 °C at the rate of 0.03 °C/min, which was controlled by the cryostat microprocessor.

### 4.3. Loading and Release of Antimicrobial Drugs into and from the PVACGs

The samples of PVA cryogels of cylindrical shape (15 × 10 mm) (Figure 2a) or the 3 mm thick discs (Figure 2b,c) were immersed into the 0.2M solution of CFT prepared using the injection quality water. The volumetric ratio of PVACG: liquid was equal to 1:5. The samples were kept at room temperature for 72 h with periodical gentle stirring to reach the equilibrium concentration of CFT; this time was found experimentally in the preliminary tests. In the case of cryogel samples intended for the in vivo experiments, the FNZ additives in an amount of 3.5 mg/mL were introduced in the loading solutions as the antifungal agent.

The release of CFT from the drug-loaded PVACG samples was studied as follows:

Each CFT-containing cylindrical cryogel sample was placed into the glass vial with 10 mL of deionized water and incubated at room temperature with periodical gentle stirring. The 0.1 mL aliquots were taken after definite time intervals, and the same pure water aliquots were returned to restore the initial volume of the liquid phase. Each 0.1 mL aliquot was diluted with 2.9 mL of water followed by the measurement of the optical absorption at 240 nm of this solution using a T70 UV/VIS Spectrometer (PG Instruments Ltd., Wibtoft, UK). Then, the respective values of the CFT concentration were found from the preliminary obtained calibration graph.

### 4.4. Physical Properties of PVACGs

The size parameters of the cylindrical cryogels, i.e., their diameter and height, were measured with a caliper, thus allowing us to calculate the volume (Vs) of the samples. The values of their Young’s compression modulus (Ec) were determined analogously to the earlier reported procedure [54,55,61] using a TA-Plus automatic texture analyzer (Lloyd Instruments, West Sussex, UK). The uniaxial loading rate was 0.3 mm/min and the tests were accomplished until a 30% deformation of the respective PVACGs. Three parallel samples were examined in three independent experiments. The obtained results were averaged.

Fusion temperatures (***T*_f_**) of the PVA cryogels were measured as follows: The cylindrical sample of either the drug-free or the CFT-loaded PVACG was placed in the plastic testtube, then a small incision was made using a scalpel in the top surface of a gel, and the stainless steel ball of 3.5 mm in diameter and 0.275 ± 0.005 g in weight was inserted in the incision. The test tube was corked and installed into the water bath. The bath temperature was increased at a rate of 0.4 ± 0.1 °C/min. The gel fusion point was detected as the temperature when the ball fell down onto the bottom of the test tube passing through the melted gel. The *T*_f_ values were measured for three parallel samples; the samples were examined in three independent experiments. The obtained results were averaged.

### 4.5. Characterization of Antibacterial Activity of the CFT-Containing PVACGs

The evaluation of the antibacterial activity of the PVA cryogel samples loaded with CFT was performed by the standard agar diffusion procedure [100]. The experiments were carried out using the 3 test bacterial strains: *Staphylococcus aureus* (Gram-positive), *Escherichia coli* (Gram-negative), and *Pseudomonas fluorescens* (Gram-negative)—all taken from the collection of the Microbiology Department, Biology Faculty of M.V.Lomonosov Moscow State University. Standard Petri dishes that contained agar nutrient medium (thickness of the medium layer was 4–5 mm) were dried at 37 °C for 40 min. The measurements of the growth inhibition zones (**GIZ**) for these bacterial strains were performed after 24 h, 48 h, and 72 h of incubation at 30 °C. Statistically reliable results were obtained by a 5-fold repetition of the measurements for each series of samples.

### 4.6. In Vivo Experiments

Chinchilla rabbits were the males of 3 kg in weight (n = 15). The manipulations did not cause pain to the animals and were carried out in compliance with the Russian legislation: GOST 33215-2014 (Guidelines for accommodation and care of laboratory animals. Rules for equipment of premises and organization of procedures) and GOST 33216-2014 (Guidelines for accommodation and care of laboratory animals. Rules for the accommodation and care of laboratory rodents and rabbits). The work was approved by the local Ethics Committee at the I.M.Sechenov First Moscow State Medical University (Sechenov University) (7 July 2021, Protocol No. 12-21).

Rabbits were under Zoletil 100 anesthesia injected intramuscularly into the thigh in an amount of 1 mL of the 4.5% solution per a rabbit. Next, a surgical scalpel was used to make an incision on the skin in the thigh area, and a cryogel-based sample was inserted subcutaneously. When it was required, bacterial infection (~20 mg weight small piece of rabbit droppings that are known to contain the consortium of such microorganisms as *Lactobacillus*, *Bifidobacterium*, *Bacillus subtilis*, *Fuzobakterium*, *Eschericia coli*, *Staphylococcus*, and some others [101]) was introduced into the wound together with an implant. Then, the wound was sewn up in layers and treated with iodine. During the post-surgery period, no antibiotic treatment was administrated. The duration of animal follow-up after the sample administration was 7, 14, and 30 days.

There were the following groups of the experimental animals:

Group 1—the rabbits were infected in the area of the wound in the thigh, the CFT/FNZ-loaded cryogel samples were subcutaneously injected into the wound area; the animals were removed from the experiment on day 7.

Group 2—the infected wound was sutured without the cryogel implant; the animal was withdrawn from the experiment on day 7.

Group 3—analogously to group 1, but the animals were withdrawn from the experiment on the 14th day.

Group 4—the rabbits were not infected in the wound area, the CFT/FNZ-loaded cryogel samples were subcutaneously inserted into the wound area in the thigh; the animals were removed from the experiment on the 30th day.

Group 5—analogously to group 1, but the animals were withdrawn from the experiment on the 30th day.

On the 7th, 14th, and 30th days, the rabbits were removed from the experiment with the help of ether anesthesia and embolism. Thereafter, the tissue with or without the inserted cryogel was cut out of the wound area and fixed in 4% neutral formalin. Further, the tissue fragments were poured into paraffin, thin sections of 7 μm in thickness were obtained according to the standard histological technique, then stained with hematoxylin-eosin and investigated using a DMRXA2 light microscope (Leica, Wetzlar, Germany) equipped with a digital camera DP70 (Olympus, Tokyo, Japan). On the 30th day, an additional smear with a sterile cotton swab was taken from the wound area and then an analysis was carried out for sowing microflora in accordance with the common procedure [102].

## Figures and Tables

**Figure 1 gels-09-00113-f001:**
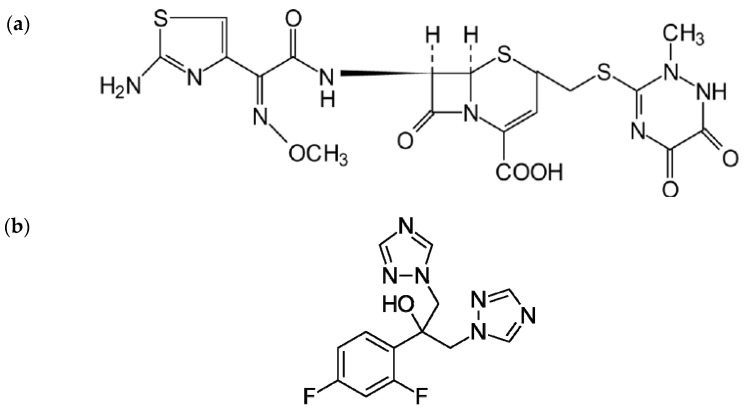
(**a**) Chemical structures of Ceftriaxone [89] and (**b**) Fluconazole [90].

**Figure 2 gels-09-00113-f002:**
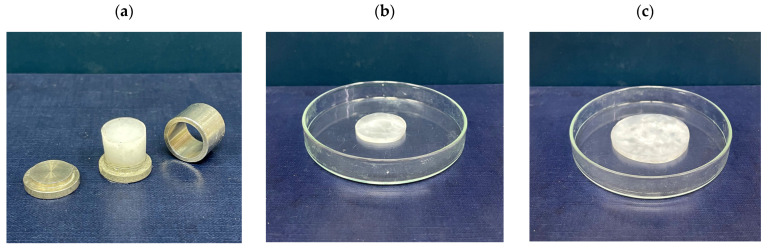
The appearance of the PVACG samples molded in the shapes of a cylinder (**a**) and the 3-mm-thick discs of different diameter (**b**,**c**).

**Figure 3 gels-09-00113-f003:**
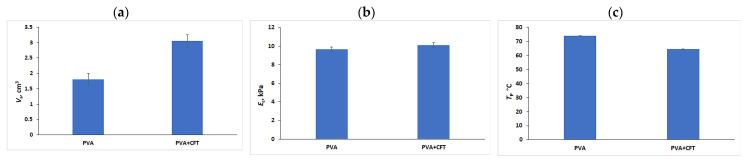
The values of the volume (**a**), compression Young’s modulus (**b**), and fusion temperature (**c**) of the PVA cryogels prior to and after loading the samples with CFT.

**Figure 4 gels-09-00113-f004:**
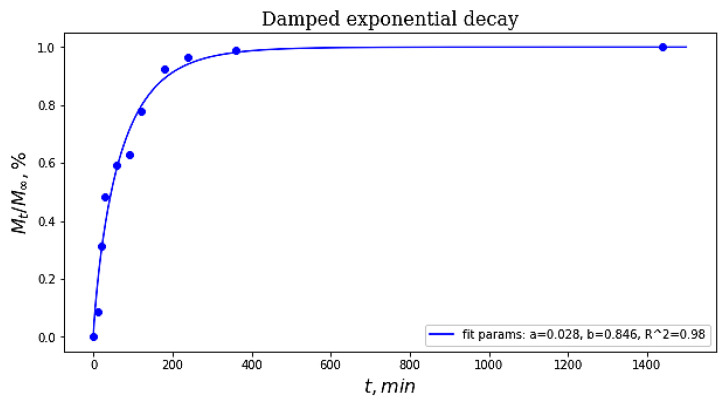
Kinetic profile of CFT release from the drug-loaded PVACG.

**Figure 5 gels-09-00113-f005:**
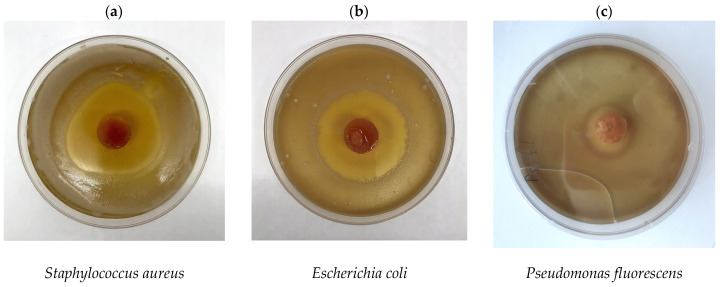
Growth inhibition zones for three strains of bacterial cells around the CFT-loaded PVACG-based carrier after 48 h of incubation.

**Figure 6 gels-09-00113-f006:**
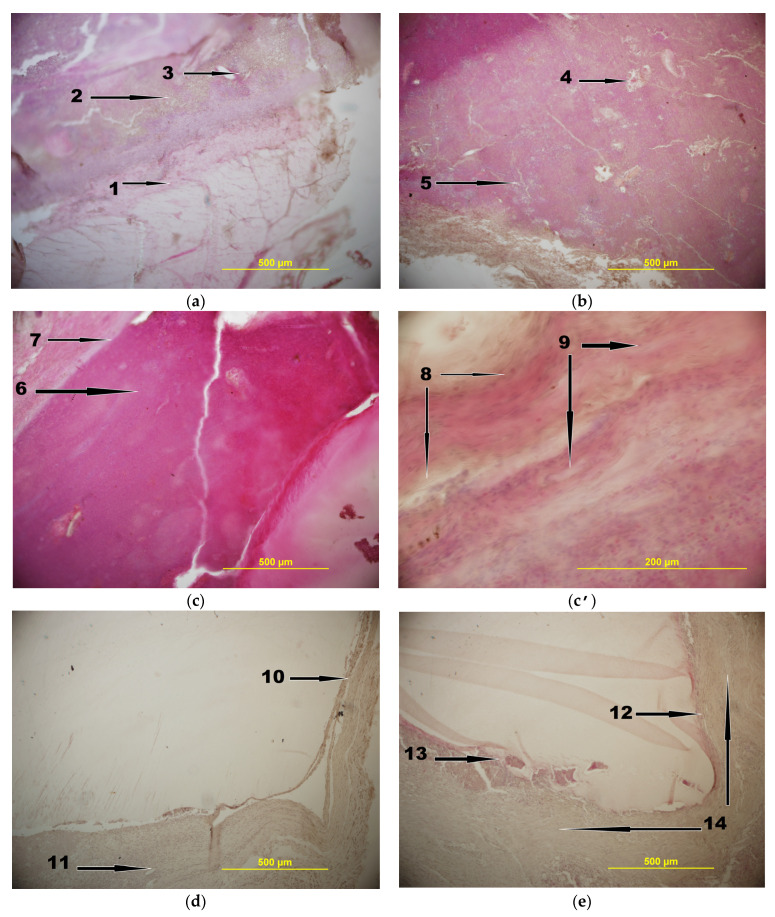
Histological sections of rabbit tissues of the experimental groups: (**a**)—group 1; (**b**)—group 2; (**c**)—group 3, low magnification; (**c’**)—group 3, a higher magnification; (**d**)—group 4; (**e**)—group 5 (for denotations and explanations see the text above this figure).

**Table 1 gels-09-00113-t001:** Growth inhibition zones for three bacterial layers around the CFT-loaded PVACGs **^(a)^**.

Bacterial	GIZ Diameter (mm) after Cell Cultivation for Definite Time		
Strain	24 h	48 h	72 h
*Staphylococcus aureus*	47 ± 2	52 ± 2	65 ± 2
*Escherichia coli*	48 ± 2	48 ± 2	44 ± 2
*Pseudomonas fluorescens*	18 ± 1	22 ± 1	25 ± 1

**^(a)^** GIZ diameters in the case of reference drug-free PVACG were zero for all tested bacteria.

## Data Availability

The data presented in this study are available on request from the corresponding author.

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
