# Peer review of "Cryostructuring of Polymeric Systems: 64. Preparation and Properties of Poly(vinyl alcohol)-Based Cryogels Loaded with Antimicrobial Drugs and Assessment of the Potential of Such Gel Materials to Perform as Gel Implants for the Treatment of Infected Wounds†"

_gels, 2023, doi:10.3390/gels9020113_

Round 1

Reviewer 1 Report

The present manuscript describes the preparation of macroporous poly(vinyl alcohol)-based cryogels, loaded with antibiotic and antimycotic agents, for biomedical application. The work can be of interest to the Readers of Gels and can be considered for the publication after addressing the following comments.

1. The Authors mention in the Introduction that there are multiple works, reporting PVA cryogels, loaded with various antibiotics. I believe it should be stressed more, why it is important to add one more antibiotic agent to the list. Was an antibiotic/antifungal agent combination in the cryogels reported before?

2. The statement about macroporous structure of the prepared cryogels should be supported by experimental data, for example, SEM images.

3. The results, presented in the work, refer mostly to antimicrobial activity of the prepared CFT/FNZ cryogels, which is determined by the CFT presence. At the same time, the effect of FNZ on the cryogel antimycotic activity and performance as the bioimplant is completely omitted. The Authors should elaborate on this part.

Author Response

Reviewer-1

The present manuscript describes the preparation of macroporous poly(vinyl alcohol)-based cryogels, loaded with antibiotic and antimycotic agents, for biomedical application. The work can be of interest to the Readers of Gels and can be considered for the publication after addressing the following comments.

  1. The Authors mention in the Introduction that there are multiple works, reporting PVA cryogels, loaded with various antibiotics. I believe it should be stressed more, why it is important to add one more antibiotic agent to the list. Was an antibiotic/antifungal agent combination in the cryogels reported before?

Replay:

In the papers related to the studies of the antibiotic-carrying PVA cryogels (refs. [84-87] in the submitted manuscript) the respective medications have been introduced in the initial aqueous PVA solutions prior to the cryotropic gel-formation, i.e. in accordance to the way (i) discussed in the lines 108-110 of the submitted manuscript. Therefore, the following phase was added in page 4 to the revised manuscript “It is this way for the preparation of the antibiotic-carrying PVA cryogels was used in the respective studies [84-87] mentioned in ‘Introduction’.”

On contrast, in the present research we used the way (ii) (lines 111-114) for CFT uploading in the already formed PVA cryogel. In this case the Ceftriaxone has been chosen as the model drug (lines 125-126) in order to demonstrate the possibility to prepare the gel implants that can contain the substances with expressed chaotropic ability (lines 141-149), since it is known that introduction of chaotropes in the aqueous PVA solutions prior to their freeze-thaw processing inhibits the formation of PVA cryogels ([61] in the submitted manuscript).

The information on the PVA cryogels that contained the antibiotic/antifungal agent combination did not meet in the scientific publications by the authors of the present paper.

  1. The statement about macroporous structure of the prepared cryogels should be supported by experimental data, for example, SEM images.

Replay:

            The macroporous morphology of PVA cryogels is a well-known fact, which has been explored many-many times using different experimental techniques and described in numerous reviews and experimental papers (lines 40-43 and refs. [1-10] in the initially submitted manuscript). In the present study, PVA cryogels “were prepared essentially in accordance to the earlier published procedure [54,55,61]” (343-344); the properties and the macroporous structure of thus formed gel matrices has been earlier studied in details [54,55]. Therefore, we think that no any necessity to add some novel experimental data (SEM images, in particular) in the present manuscript. The readers who are interested in the already published information about the features of macroporous structure of similar PVA cryogels can simply find such data in the cited references.

            With respect of influence of the gel-formation parameters on the macroporous structure of PVA cryogels we added the following note in page 2 of the revised manuscript: “With that the influence of these variables on the macroporous morphology of the resultant PVA cryogels has also been explored and reported in details [3,5-10,47,48,53-55,60-63].”

  1. The results, presented in the work, refer mostly to antimicrobial activity of the prepared CFT/FNZ cryogels, which is determined by the CFT presence. At the same time, the effect of FNZ on the cryogel antimycotic activity and performance as the bioimplant is completely omitted. The Authors should elaborate on this part.

Replay:

            In this study, the antifungal agent Fluconazole was applied for the sake of antifungal defense in the case of in vivo experiments, when the drug-loaded PVA cryogel samples were used. The FNZ additives in an amount of 3.5 mg/mL (or ~0.011 mol/L) were introduced in the loading solutions (this concentration is recommended by manufacturer of the drug). At the same time, the CFT concentration in the loading medium was 0.2 mol/L (section 4.3 in the submitted manuscript), i.e. higher by a factor of ~18. Therefore, we studied only the CFT influence on the physico-chemical characteristics of the drug-loaded PVA cryogels. In turn, in all in vivo experiments, when the CFT/FNZ-loaded implants were used, no exhibition of fungal infection was detected in the microbiological tests, whereas the mycotic problems took place for the reference infected rabbits that did not have the implants. Such results confirmed the antifugal action of FNZ released from the CFT/FNZ-containing implants.

We added the following note in page 10 of the revised manuscript: “With that, no fungal contamination was found, thus indicating the antimycotic efficacy of the FNZ additives released from such implants, whereas the smears from the wounds of infected rabbits without the implants (reference group 2) contained fungal admixtures.”

In the framework of this particular study, there was no necessity to perform some separate studies with respect of the antimycotic activity for the implants that contained FNZ only.

Reviewer 2 Report

In the study, the authors examined the use of antimicrobial and antifungal drug-absorbed poly(vinyl alcohol) cryogels as implants in infected wounds. The manuscrispt, in which the results of in vivo and in vitro experiments are shared, is written in a very clear language.

You can find my detailed opinions and suggestions about the study below.

Minor revisions:

ü  None of the parameters that determine the properties of the PVA-based freeze-gammon cryogels, as stated by the authors (lines 48-52), were used in the study. The authors worked at a single molecular weight and concentration, a single freeze-dry cycle, and a single freeze temperature. Although the authors' main focus in the study was gel implants containing Ceftriaxone and Fluconazole drugs, the possible effects should be briefly shared in the text and in the results section. I recommend that authors can cite recent publications examining cryogelation conditions and final product properties.

ü  Section 2.1, lines 153-164: The authors interpreted the decrease in fusion temperature of CFT-loaded cryogels as the breakdown of interchain hydrogen bonds due to the chaotropic effect of CFT. However, they could not see this effect in Young's modulus, which is directly related to the crosslink density in the material. Moreover, the cross-links, which are reduced by the effect of CFT, increase swelling. What about solubility and gel fraction?

ü  The authors used 3 different variants in in vivo test. The first is time: Groups 1, 3 and 5 show 7th, 14th, and 30th days of wound healing in the presence of drug-loaded cryogel implants, respectively. Second variant drug-loaded implant: Shows 7th day results in the presence of Group 1 implant while Group 2 has no implant. The last parameter is infection: it shows the 30-day healing results of Group 4 uninfected wound and Group 5 infected wound in the presence of drug-loaded cryogel implant. Despite this detailed study, the results were described roughly and independently of the control groups. I suggest the authors explain the range of lines 230-270 with variables in mind.

Author Response

Reviewer-2

In the study, the authors examined the use of antimicrobial and antifungal drug-absorbed poly(vinyl alcohol) cryogels as implants in infected wounds. The manuscrispt, in which the results of in vivo and in vitro experiments are shared, is written in a very clear language.

  1. None of the parameters that determine the properties of the PVA-based freeze-gammon cryogels, as stated by the authors (lines 48-52), were used in the study. The authors worked at a single molecular weight and concentration, a single freeze-dry cycle, and a single freeze temperature. Although the authors' main focus in the study was gel implants containing Ceftriaxone and Fluconazole drugs, the possible effects should be briefly shared in the text and in the results section. I recommend that authors can cite recent publications examining cryogelation conditions and final product properties.

            Replay:

            First of all, we would like to point out that the term “freeze-gammon cryogels” used by this reviewer is unknown in the scientific literature related to the cryogels and the cryotropic gel-formation. Is it, may be, some laboratory slang?

            Second, the PVA used in our study (Mowiol® 28-99, Merck) was selected because the polymer of this particular trade mark was approved for the biomedical implementation (added in the ‘Materials’ section (page 11) of the revised manuscript).

            The studies of the interrelations between the “cryogelation conditions and final product properties” were not the goals of this our research, therefore in the ‘Introduction’ we gave several key references (e.g. [1-10,45-67], including recent (2022) review by Adelnia et al. [10]). Generally, these publications are commonly known and were quoted many times in the respective literature. We believe that even a brief discussion of the dependences of cryogels’ properties on the characteristic of the gel-forming polymer and the conditions of cryogenic processing will only overload the text of the paper by the well-known information, which is not related directly to the subject of the present study. The readers who are interested in the already published information about these interrelations can simply find such data in the cited references.

  1. Section 2.1, lines 153-164: The authors interpreted the decrease in fusion temperature of CFT-loaded cryogels as the breakdown of interchain hydrogen bonds due to the chaotropic effect of CFT. However, they could not see this effect in Young's modulus, which is directly related to the crosslink density in the material. Moreover, the cross-links, which are reduced by the effect of CFT, increase swelling. What about solubility and gel fraction?

            Replay:

            The idea that the up-swelling pressure can participate in the maintance of gel strength after loading of PVA cryogel with the choatrope-like CFT has been formulated in the initial version of our manuscript as an assumtion: “Such results can testify to a positive contribution of the up-swelling pressure to the elasticity characteristics of the CFT-loaded PVACGs (Fig. 3a), thus compensating some swelling-induced softening of the gel matrix.” Of course, additional studies are required to confirm or to refute this assumption. However, similar studies were not included in the objectives of our present research. The last two sentenses have been added to the revised manuscript in page 5.

  1. The authors used 3 different variants in in vivo test. The first is time: Groups 1, 3 and 5 show 7th, 14th, and 30th days of wound healing in the presence of drug-loaded cryogel implants, respectively. Second variant drug-loaded implant: Shows 7th day results in the presence of Group 1 implant while Group 2 has no implant. The last parameter is infection: it shows the 30-day healing results of Group 4 uninfected wound and Group 5 infected wound in the presence of drug-loaded cryogel implant. Despite this detailed study, the results were described roughly and independently of the control groups. I suggest the authors explain the range of lines 230-270 with variables in mind.

            Replay:

As for the comparison of data for the animals of groups 3 and 5 with the animals of control group 2 (the infected rabbits without the drug-loaded implants withdrawn from the experiment on day 7), similar control rabbits died before the 14th day as a consequence of a strong infectious effect. Therefore, for the 14- and 30-days animals of groups 3 and 5 (initially infected rabbits with inserted drug-containing implants), comparison with the control was not possible. The corresponding explanation was included in the text of the revised manuscript in page 8.

In turn, the animals of group 4 (uninfected rabbits with implanted drug-loaded cryogels) were needed to find out whether a foreign body in the form of such an implant causes any inflammatory process in the wound. It turned out that there was no inflammation process in this case: the implant was simply encapsulated with a fibrous capsule without inflammatory processes and any changes in the surrounding tissues.

Round 2

Reviewer 1 Report

The Authors have addressed the previous remarks. The manuscript can be accepted.